# Platelet Aggregation, Mitochondrial Function and Morphology in Cold Storage: Impact of Resveratrol and Cytochrome c Supplementation

**DOI:** 10.3390/cells12010166

**Published:** 2022-12-30

**Authors:** Michael L. Ekaney, Juan Carlos Carrillo-Garcia, Gabrielle Gonzalez-Gray, Hadley H. Wilson, Mary M. Jordan, Iain H. McKillop, Susan L. Evans

**Affiliations:** FH “Sammy” Ross Trauma Center, Department of Surgery, Carolinas Medical Center, 1000 Blythe Boulevard, Charlotte, NC 28203, USA

**Keywords:** platelet storage, resveratrol, cytochrome c, platelet aggregation, platelet morphology, mitochondrial function

## Abstract

Donated platelets are critical components of hemostasis management. Extending platelet storage beyond the recommended guidelines (5 days, 22 °C) is of clinical significance. Platelet coagulation function can be prolonged with resveratrol (Res) or cytochrome c (Cyt c) at 4 °C. We hypothesized that storage under these conditions is associated with maintained aggregation function, decreased reactive oxygen species (ROS) production, increased mitochondrial respiratory function, and preserved morphology. Donated platelets were stored at 22 °C or 4 °C supplemented with 50 μM Res or 100 μM Cyt c and assayed on days 0 (baseline), 5, 7 and 10 for platelet aggregation, morphology, intracellular ROS, and mitochondrial function. Declining platelet function and increased intracellular ROS were maintained by Res and Cyt c. Platelet respiratory control ratio declined during storage using complex I + II (CI + CII) or CIV substrates. No temperature-dependent differences (4 °C versus 22 °C) in respiratory function were observed. Altered platelet morphology was observed after 7 days at 22 °C, effects that were blunted at 4 °C independent of exposure to Res or Cyt c. Storage of platelets at 4 °C with Res and Cyt c modulates ROS generation and platelet structural integrity.

## 1. Introduction

Platelet transfusion is an established practice to facilitate blood clotting and prevent excessive bleeding in the setting of trauma [1,2,3,4], while prophylactic platelet transfusions are commonly performed to treat secondary thrombocytopenia and prevent hemorrhage in cancer patients [5,6]. While single donor apheresis platelets are considered standard of care in many centers in Europe, between one-quarter and one-third of all transfusions in the United States occur using pooled platelets [7].

Current Food and Drug Administration guidelines for platelet storage recommend use within 5 days of collection when platelets are stored at 22 °C due to the increased risk of bacterial contamination. This relatively short “shelf life” (compared to other blood products) means approximately 30% of platelet concentrates expire prior to being used in transfusions, contributing to an ongoing shortfall in availability [8]. Innovative strategies to minimize platelet waste and increase availability are of direct clinical relevance [9,10].

Previous studies report that mitochondrial dysfunction in platelets contributes to diminishing functional capacity during storage [11]. Dysfunctional mitochondria are characterized by reduced respiratory capacity and decreased adenosine triphosphate (ATP) production [12,13,14]. The most consistent defect driving abnormal mitochondrial function is impairment of mitochondrial electron transport chain (ETC) complexes [15]. During normal ATP production, mitochondria generate reactive oxygen species (ROS) which can actively contribute to intracellular signaling [16]. However, pathological states and conditions of cell stress can lead to increased and unregulated mitochondrial ROS production that overwhelms endogenous antioxidant systems. Under these conditions, intracellular oxidative stress contributes to irreparable changes in mitochondrial protein structure, morphology, and function [17].

During platelet storage, elevated intracellular ROS is associated with declining mitochondrial function [11]. In an effort to counteract the effects of ROS or improve ETC efficacy, we and others have examined stored platelet concentrates supplemented with resveratrol (Res; a polyphenolic antioxidant) or cytochrome c (Cyt c; an electron carrier of the ETC) [18,19]. Other polyphenolic compounds, having similar properties to Res, exist (e.g., quercetin and kaempferol [20,21]). The focus on Res in this study was based on its potential adjunct use in resuscitative therapy for hemorrhagic shock, a condition characterized by mitochondrial dysfunction, tissue hypoxia, and inflammation [22,23,24]. In addition, Res is reported to interact with other potential targets to preserve organ function in rodent models in the absence of toxicity [25]. Equally, Cyt c (an electron carrier in the ETC) is released into the circulation under conditions of stress and modulates a cascade of cellular responses, including apoptosis [26,27,28]. Platelets stored at 4 °C exhibit extended aggregation function beyond the recommended 5 day storage shelf-life, and combining 4 °C storage with Cyt c supplementation further maintains aggregation function compared to cold storage alone [29,30].

Understanding the mechanisms involved in platelet dysfunction may provide further insight for identifying targets to improve therapeutic preservation and correction of platelet dysfunction during storage and increase clinical availability. The goals of this study were to elucidate mechanisms by which cold storage of platelets supplemented with Res or Cyt c maintains platelet function [29]. We hypothesized that preserved aggregation function in cold storage is associated with decreased platelet mitochondrial ROS production, increased respiratory function, and preserved structural integrity following Res and Cyt c supplementation in platelets stored under hypothermic conditions.

## 2. Materials and Methods

### 2.1. Assurances

The institutional review boards of the Carolinas Medical Center and OneBlood approved this study. Written informed consent was obtained from all donors at OneBlood.

### 2.2. Platelet Collection and Preparation

Whole blood was collected in citrate phosphate dextrose (CPD) solution (70 mL CPD/500 mL unit of whole blood) from 20 healthy volunteers (OneBlood, Charlotte, NC, USA). From these 20 individual collections, platelet rich plasma (PRP) was obtained by centrifugation (10 min at 170× *g*) and pooled (4 pooled PRP samples, 5 donors per pooled sample). Each set of pooled PRP samples (*n* = 4) was then aliquoted equally into individual platelet-storage bags (Haemonetics, Braintree, MA, USA) in 100% plasma using an aseptic technique. Pooled PRP samples were randomly assigned to 4 °C or room temperature storage and designated as control (vehicle; 1X phosphate-buffered saline [PBS]), Res (50 μM), or Cyt c (100 μM)).

To equalize platelet concentration within the experimental bags an aliquot of PRPs was diluted 1/5000 (ISOTON II diluent) and platelet number determined using a z2 Coulter Counter (Beckman Coulter, Brea, CA, USA) with particle size set to 1.3–2.4 μm. Final platelet concentration was adjusted to 1400 × 10^3^ cells/μL and a total volume of 45 mL per (150 mL) pedi-pack storage bag was used. Doses of Res and Cyt c were based on previous data reporting increased coagulation function without observed toxicity in vivo [29]. Platelet concentrates stored at 22 °C were subject to mild agitation (70 rpm) in accordance with recommended storage conditions for clinical application. Platelets stored at 4 °C did not undergo agitation as previous studies report agitation does not improve platelet function [31]. Aliquots (4 mLs) were removed and assayed on days 0, 5, 7 and 10 via a sterile port under aseptic conditions.

### 2.3. Light Transmission Aggregometry

Following sample collection (4 mLs/time point), samples were divided, one aliquot was subjected to centrifugation (10 min at 2500× *g*), and the supernatant collected for use as the plasma poor platelet (PPP) sample. Light transmission aggregometry was performed using a two-channel Chronolog 700 Aggregometer (Chronolog Corporation, Havertown, PA, USA). Following preliminary analyses using 250,000 platelets/μL, an unstable optical baseline was observed. To correct for this platelet concentration of PRP, samples were adjusted using PPP to a final concentration of 500,000 cells/μL with 500 μL of adjusted PRP used for analysis. Platelet aggregation was induced using 5μL adenosine diphosphate (ADP, final concentration 10 μM, CHRONO-PAR^®^ ADP, Chronolog Corporation) or 5 μL arachidonic acid (AA, 0.5 mM final concentration, CHRONO-PAR^®^ Arachidonic Acid; Chronolog Corporation). An amount of 500 μL of PPP was used as a control representing 100% light transmission or 100% aggregation, while PRP represented 0% light transmission or 0% aggregation prior to stimulation. The aggregation tracing was run for a minimum of 6 min or when tracings reached full amplitude after agonist addition (ADP or AA). Results are represented as % ADP or AA aggregation representing maximum amplitude after stimulation.

### 2.4. Oxidative Stress in Platelet Concentrates

Total ROS production was quantified using a 2′,7′-dichlorofluroescein diacetate (DCF) dye (Sigma-Aldrich, St. Louis, MO, USA) [32]. Platelet concentrates were diluted 7-fold in 1X PBS, and DCF was added to a final concentration of 20 µM. Samples were incubated for 30 min (37 °C) and fluorescence intensity was measured by spectrophotometry (excitation/emission 504/529 nm).

### 2.5. Platelet Mitochondrial Respiration

Platelet number was determined in sample aliquots using a z2 Coulter Counter as previously described. Platelets were diluted 1:6 in 1X PBS. The respiratory capacity of the different ETC complexes was determined by measuring oxygen consumption over time in an enclosed chamber using a 782 Strathkelvin Oxygen Meter (Strathkelvin Instruments, North Lanarkshire, UK).

To analyze specific components of platelet mitochondrial respiratory capacity, we employed selective substrates and/or inhibitors of the complexes of the mitochondrial ETC. Briefly, substrates and/or inhibitors of Complex I (substrate-glutamate (5 mM) + substrate-malate (2.5 mM); Complex II (substrate-succinate (5 mM) + CI inhibitor-rotenone (2 μM); (Complex I+II (substrate-glutamate (5 mM) + substrate-malate (2.5 mM) + substrate-succinate (5 mM)); Complex IV (substrate-ascorbate (6 mM) + substrate-TMPD (300 μM) + inhibitor-antimycin A of CIII (250 nM)) were added to diluted platelets and mixed by repeat pipetting [33]. Aliquots (60 µL) were loaded into an enclosed respirometer and allowed to equilibrate for 2 min prior to the addition of ADP (250 µM), after which analysis of oxygen consumption was performed. The respiratory control ratio (RCR) was calculated whereby an RCR of 1 indicates dysfunction in the respiratory capacity of the targeted complex associated with ADP-coupled respiration and is the minimum value possible. Five minutes after adding ADP, an ETC uncoupler (carbonilcyanide p-triflouromethoxyphenylhydrazone [FCCP, 200 nM]) was added and oxygen consumption was measured to determine the uncoupled respiration rate. Data are presented as a pattern of change in RCR over time as it is invalid to perform statistical analysis when the RCR value = 1 (i.e., the baseline that indicates respiration is no longer occurring).

### 2.6. Transmission Electron Microscopy (TEM)

Platelets were fixed in Karnovsky’s fixative solution and processed in a Pelco Biowave Pro laboratory microwave system for 1 hour using cacodylate buffer rinses, 1% osmium tetroxide, and ethanol. Processed platelets were placed in 100% spur resin, placed on 200 mesh thick copper grids, and stained with uranyl acetate and lead citrate. Sections were viewed and images captured using a Jem 1400 Plus transmission electron microscope (JEOL Ltd., Peabody, MA, USA). Platelet integrity was blind scored using a pre-defined scoring scale of 0–3 where 0 represents healthy platelets and 3 represents severely damaged platelets. In total, 20 images from each storage condition (Day 0 and Day 7, 22 °C; Day 7, 4 °C; Day 7, 4 °C + Cyt c, and Day 7, 4 °C + Res) were captured and blind scored with data presented as mean score ± SEM.

### 2.7. Statistical Analysis

Statistical analysis was performed using GraphPad Prism software v9.0 (GraphPad Software, San Diego, CA, USA). Data are presented as Mean ± SEM and analyses were performed by applying a one-way analysis of variance (ANOVA) and Tukey’s post hoc test. Using this method, data were analyzed within groups to determine change compared to day 0 (baseline) for each condition, or between groups to determine the effect of experimental condition versus control (no treatment) for the same time point. A *p*-value < 0.05 was considered significant.

## 3. Results

### 3.1. Light Transmission Aggregation

ADP-dependent platelet aggregation decreased during storage in control platelets on day 5, 7, and 10 versus day 0 in both 22 °C and 4 °C storage, with the extent of the decrease being more prominent at 22 °C compared to 4 °C (Figure 1A,B). At 4 °C, storage Res and Cyt c significantly preserved platelet function on day 7 and day 10 versus control day 7 and day 10, respectively (Figure 1B). AA-dependent platelet aggregation remained unchanged on day 5 and day 7 but significantly declined at day 10 versus day 0 at 22 °C (* *p* < 0.05). At 4 °C, AA-dependent aggregation remained unchanged from day 0 to day 15.

### 3.2. Oxidative Stress in Platelet Concentrates

Cytoplasmic ROS levels in the control (vehicle only) group increased following storage at 5, 7 and 10 days versus day 0 at both 22 °C and 4 °C (Figure 2A,B, * *p* < 0.05 versus day 0, *n* = 5). Supplementation with Res or Cyt c led to lower ROS levels in stored platelets compared to vehicle-treated platelets at pair-matched storage time points (Figure 2A,B, # *p* < 0.05 Res or Cyt c versus pair-matched control time point, *n* = 5).

### 3.3. Respirometry in Platelet Concentrates

All samples on day 0 of storage, independent of supplementation or storage temperature, had similar initial RCRs. (Figure 3A–F). Using complex I (CI) and complex II (CII) dependent substrates, RCR decreased from day 0 to day 5 at 22 °C and 4 °C (Figure 3A,B). Furthermore, respiratory capacity in the control group attained minimum capacity (RCR = 1.0) by day 5 (Figure 3A,B). In contrast, supplementation with Res or Cyt c maintained an RCR > 1.0 on day 5. Although RCR is maintained >1, there was no statistical difference between control and Res or Cyt c conditions at day 5 (Figure 3A,B). When using a CII selective substrate, RCR did not change significantly over time and was not different between groups, independent of either Res or Cyt c supplementation or temperature (Figure 3C,D). Using complex IV (CIV) dependent substrates, a gradual decrease in RCR was measured with increasing storage time at both 22 °C and 4 °C, with no apparent effect of supplementation with Res or Cyt c (Figure 3E,F).

### 3.4. Transmission Electron Microscopy (TEM) of Platelets

Representative images (*n* = 20/experimental group) were captured and blind scored for cellular integrity (Figure 4A). Platelets stored for 7 days at 22 °C demonstrated the highest level of loss of integrity compared to platelets (Figure 4B–K). Structural integrity was better preserved at 4 °C compared to 22 °C, but no difference was observed in platelets supplemented with Res or Cyt c (Figure 4B–K).

## 4. Discussion

The goal of this study was to address potential mechanisms by which 4 °C storage and supplementation with either Res or Cyt c maintains aggregation function of stored platelets [29] compared to current FDA guidelines (storage at 22–24 °C). Here, we corroborate previous reports of an observed decline in platelet aggregation during 10-day storage at 22 °C, while cold storage without agitation preserved platelet aggregation function. The decision to use platelets stored at 4 °C without agitation arises from previous reports demonstrating no functional differences between agitated and unagitated cold stored platelets [34]. Nonetheless, the use of agitation during cold storage may impact other components of platelet function. For example, a recent study by Shea and colleagues reports that when comparing agitated and unagitated platelets at 4 °C, agitation resulted in macroaggregation while absence of agitation resulted in film-like sedimentation. However, these effects were observed in the absence of changes in surface receptor expression (CD42b, CD49, CD62P, CD63) [35]. Conversely, it is indicated that agitation of platelets stored at room temperatue is directly relevant for oxygenation purposes to prevent a rapid fall in pH due to changes in metabolic rate [34]. Most importantly, we demonstrated that Res and Cyt c supplementation maintained ADP-induced platelet aggregation at 4 °C storage. In this study, we further observed that a decline in platelet aggregation during storage is associated with increased ROS levels throughout storage time under standard conditions. This increase was reversed by supplementation with Res or Cyt c. Using RCR as a measure of respiratory function, we report RCR decreased with storage time when CI and CIV dependent substrates were used. Although untreated platelets became completely dysfunctional by day 5, the decline in respiratory function was slowed by Res or Cyt c supplementation. Finally, we demonstrated that cold storage preserves platelet structural integrity compared to 22 °C after 7 days of storage.

Reactive oxygen species (ROS) are generated within activated platelets and play an important role in regulating platelet responses to collagen and collagen-mediated thrombus formation [36]. Increased intracellular ROS is associated with oxidative stress and, if left unchecked, can lead to critical damage to a variety of biological targets that underlie disease development and progression [37]. In vivo, intrinsic antioxidant systems keep ROS levels in check to prevent the negative consequences of excessive intracellular ROS [38]. The isolation and storage of platelets ex vivo creates a stress environment for platelets which is demonstrated by the increased ROS levels measured during storage. ROS levels were significantly higher on day 5, 7 and 10 versus day 0 of storage, corroborating previous reports by Ghasemzadeh et al. (which was limited to a maximum of 5 days storage) [39]. An important observation associated with the increase in ROS during storage at both 22 °C versus 4 °C on day 5, day 7 and day 10 versus day 0 was that ROS levels on day 7 and 10 were lower, though not significantly so, than on day 5. Although it is expected that ROS levels be increased throughout the storage period, it remains to be elucidated why ROS levels trend to be lower on day 7 and day 10 versus day 5. While this might be possible due to exhaustion of cellular metabolic function limiting the ability to produce further ROS, this is speculation that requires further experimentation.

The use of antioxidants in the prevention and supplementation of thrombotic or cardiovascular diseases has been investigated with positive benefits reported [40]. Here, we report the beneficial effect of antioxidants in reducing ROS levels during platelet storage. This may underlie the beneficial effects, such as improved coagulation function in vitro, when Res or Cyt c is supplemented in platelets prior to storage [29].

We further observed that ROS levels increased independent of storage temperature (22 °C versus 4 °C). Previous studies using isolated mitochondria report increased ROS release in association with higher temperatures [41]. Although further investigations are needed to confirm that ROS levels are dependent on temperature, it is speculated that ROS levels are higher in isolated mitochondria than intact mitochondria within a cell. In addition, our observed increase in ROS levels could be a result of the closed bag system employed versus a dynamic circulation system.

Similar to ROS production, RCR was not affected by temperature in our study. This is in contrast with previous studies that report platelets stored at 4 °C conserved mitochondrial respiration and maximal mitochondrial utilization better than room temperature platelets at day 7 of storage [42]. However, consistent with previous findings, our study reports that loss in mitochondrial respiratory function could be mediated through CI and CIV. For example, Perales et al. report a dramatic decrease in platelet mitochondrial respiration for CI and CIV within the first 2 days of storage under standard (22 °C) conditions. [11] Similarly, Diab et al. report decreasing CIV activity during platelet storage [43]. This decreased respiratory capacity likely contributes to the decline in platelet aggregation, as platelet activation requires the energy production provided by mitochondrial respiration [12]. In addition, we assessed the effects of platelet storage on respiratory capacity of CII, finding that CII is not affected. This suggests that CII is not a target for interventions to prevent platelet dysfunction during storage.

In our study, supplementation with Res or Cyt c blunted the decline in RCR though it did not reach statistical significance. We did not examine RCR between days 1 and 5, and it is possible that repeated dosing with Res or Cyt c during the first days of storage could act to substantially slow the loss in respiratory function.

Stored platelets undergo deleterious changes, referred to as platelet storage lesions [44], which accelerates the desialylation of platelets and result in phagocytotic clearance by hepatic macrophages [45]. We observed structural changes in stored platelets using TEM. As previously reported [45], we noted alterations in plasma membranes, degranulation and formation of multiple pseudopodia, a hallmark of platelet activation. The structural features of platelets were better preserved when stored at 4 °C though the role of cold storage in preserving integrity remains to be defined. One possible explanation is that the decrease in oxidative stress (as a result of lower ROS production under hypothermic conditions) results in maintenance of structure.

In considering our data, there are important limitations that should be considered such as the introduction of artifacts when platelets are supplemented with Res and Cyt c affecting the baseline integrity of platelets, and that our study focused on the function of Res and Cyt c in vitro and not in vivo.

## 5. Conclusions

This study demonstrated that improved aggregation function during storage of platelet units at 4 °C, as well as supplementation with Res or Cyt c, is associated with decreased ROS levels, blunted mitochondrial dysfunction compared to current recommended storage conditions at 22 °C. Structural integrity, as depicted using electron micrography, was maintained at 4 °C without any significant change when supplemented with Cyt c. It is evident that the observed maintenance of aggregation function depicted by light transmission aggregometry could be associated with or affected by ROS production, mitochondrial function, and structural integrity. The function of these platelets in an in vivo system, mechanism of action, as well the potential for the synergistic effect of Res and Cyt c, is yet to be determined.

## Figures and Tables

**Figure 1 cells-12-00166-f001:**
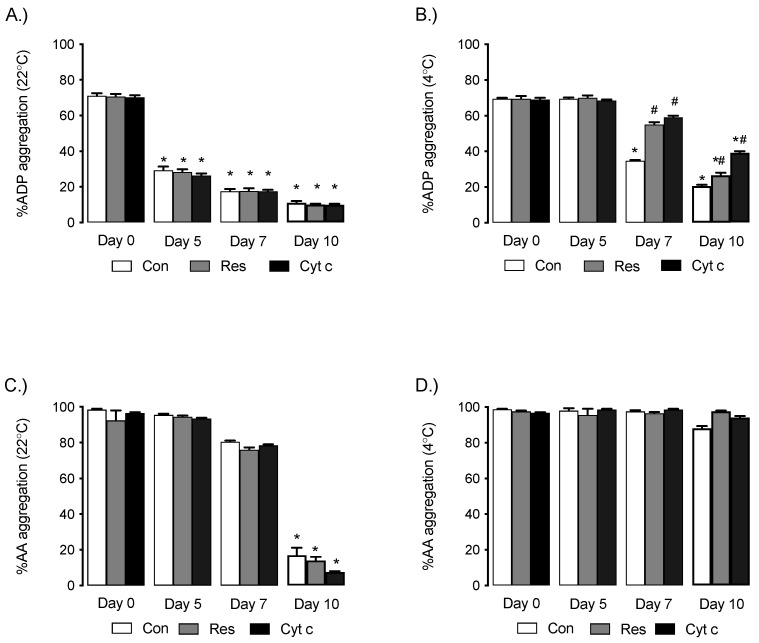
(**A**) ADP-induced platelet aggregation declines during storage in control, Cyt c and Res supplemented groups at 22 °C on day 5, day 7 and day 10 versus day 0. * *p* < 0.05 versus treatment-matched day 0, *n* = 4 independent experiments. (**B**) Decline in ADP-induced platelet aggregation is lessened by Res and Cyt c supplementation compared to untreated platelets. * *p* < 0.05 versus treatment-matched day 0, # *p* < 0.05 treatment group versus control (Con) at same time point, *n* = 4 independent experiments. (**C**) AA-induced aggregation declines during storage at 22 °C on day 10 versus day 0. * *p* < 0.05 versus treatment-matched day 0, *n* = 4 independent experiments. (**D**) AA-induced aggregation is not affected by storage at 4 °C or by supplementation with Res or Cyt c (*n* = 4 independent experiments).

**Figure 2 cells-12-00166-f002:**
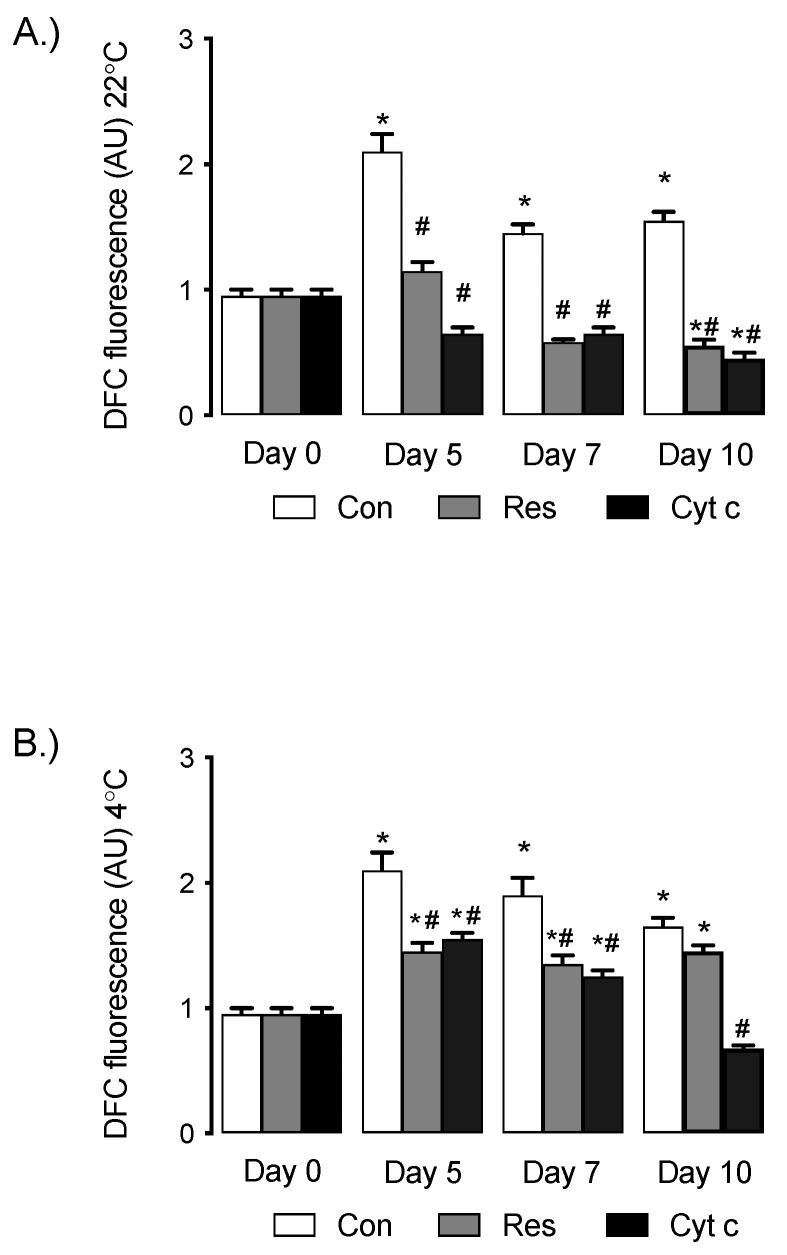
(**A**) Intracellular ROS production (measured as DFC fluorescence; arbitrary units [AU]) increases in platelets stored at 22 °C and supplementation with Res or Cyt c reduces ROS production compared to untreated (control; Con) platelets. * *p* < 0.05 versus treatment-matched day 0, # *p* < 0.05 treatment group versus Con at same time point, *n* = 4 independent experiments. (**B**) Intracellular ROS production increases in platelets stored at 4 °C, and supplementation with Res or Cyt c significantly reduces ROS production compared to Con platelets. * *p* < 0.05 versus treatment-matched day 0, # *p* < 0.05 treatment group versus Con at same time point, *n* = 4 independent experiments.

**Figure 3 cells-12-00166-f003:**
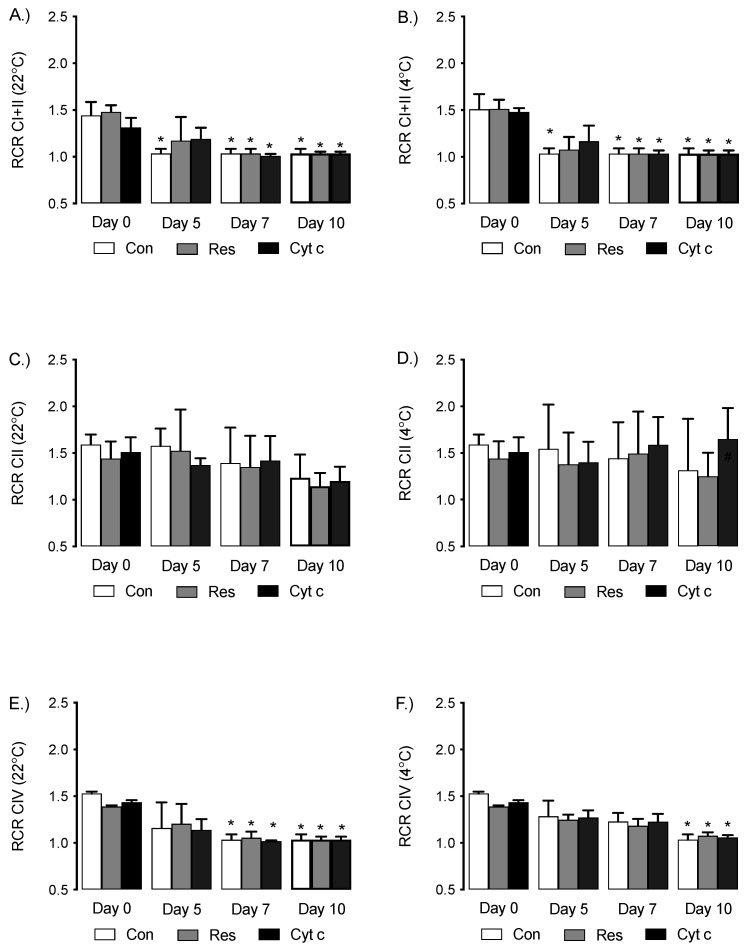
Respiratory control ratio (RCR; a measure of respiratory capacity) measured in platelets stored at 22 °C (**A**,**C**,**E**) and 4 °C (**B**,**D**,**F**). Platelets were exposed to glutamate (5 mm) + malate (2.5 mM) (CI+CII; **A**,**B**), succinate (5 mM) + rotenone (2 uM) (CII; **C**,**D**), or ascorbate (6 mM) + TMPD (300 uM) + antimycin (250 nM) (CIV; **E**,**F**). * *p* < 0.05 versus treatment-matched day 0, *n* = 4 independent experiments.

**Figure 4 cells-12-00166-f004:**
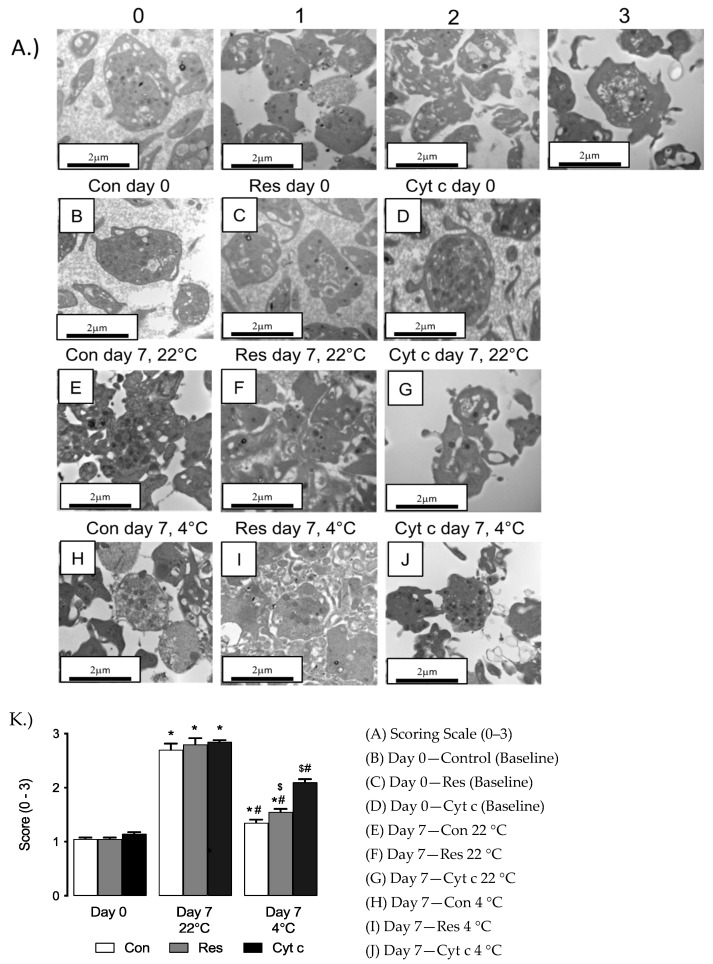
(**A**) Representative images from transmission electron microscopy (TEM; 25,000× magnification) of platelets under different storage conditions used to develop a scoring scale for morphological platelet evaluation. Scale bar = 600 nm. Representative image of platelets stored at (**B**) Untreated control (Con) day 0; (**C**) Resveratrol (Res) day 0; (**D**) Cyt c day 0; (**E**) Con day 7, 22 °C; (**F**) Res day 7, 22 °C; (**G**) Cyt c day 7, 22 °C; (**H**) Con day 7, 4 °C; (**I**) Res day 7, 4 °C; (**J**) Cyt c day 7, 4 °C. (**K**) Cold storage improves platelet structural integrity compared to 22 °C, effect of Res or Cyt c supplementation. Representative transmission electron micrographs were blind-scored (0–3 scale). * *p* < 0.05 versus treatment-matched day 0, # *p* < 0.05 4 °C day 7 versus 22 °C day 7 for same treatment group, ^$^
*p* < 0.05 Res and Cyt C supplemented cells versus untreated.

## Data Availability

Not applicable.

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
