# Peer review of "Platelet Aggregation, Mitochondrial Function and Morphology in Cold Storage: Impact of Resveratrol and Cytochrome c Supplementation"

_cells, 2022, doi:10.3390/cells12010166_

Round 1

Reviewer 1 Report (Previous Reviewer 1)

Dr Ekaney et al. resubmitted the original paper entitled „Altered Platelet Morphology and Mitochondrial Function is Reversed by Resveratrol and Cytochrome C Supplementation in Storage”. The Authors performed a series on new experiments using a method of light transmission aggregometry to assess a function of platelets stored at various conditions. Besides, the manuscript was modified according to the suggestions of three reviewers. In my opinion, the paper was significantly improved, however I have one comment.

The Authors used PRP in the light transmission aggregometry (lines 97-98) in the concentration of 500,000 cells/ul. It is high concentrations exceeding both typical concentration of PRP in aggregation studies 200,000-300,000 cells/ul, as well as physiological concentration of platelets in blood. Could the Authors explain a rationale of using this approach?

Author Response

The authors would like to thank the reviewers for providing their detailed and constructive comments for our resubmitted manuscript studying platelet morphology and function under altered storage conditions. We have now had the opportunity to carefully consider the comments and have addressed them through revisions to the manuscript body and figures. As requested, we have created two versions of the revised manuscript, one in which changes are highlighted and a second clean version. To further assist the Reviewers, we have addressed the points raised herein.

As the reviewers are aware, in the original manuscript we used TEG as an indicator of platelet function. In doing so, the reviewers highlighted that TEG was not an appropriate means of measuring platelet function, and to do so studies should be performed using aggregrometry or flow cytometry. We agreed with these comments and performed the aggregrometry studies as requested. With the resubmission we included both TEG and aggregometry data for comparison. However, as highlighted by the reviewers following this review, this causes confusion in data presentation, interpretation and discussion. To address this, we believe the data is better represented by removing the TEG data completely and only presenting data obtained by aggregrometry.

In addition, as highlighted by the reviewers, the nomenclature used to designate the time from platelet isolation to use was confusing as it suggested the platelets were kept for 1 day prior to experiments beginning. This was not the case and to address this, platelets have been designated Day 0 in the text and figures.

Reviewer 1

Major comments

Dr Ekaney et al., resubmitted the original paper entitled “Altered Platelet Morphology and Mitochondrial Function is Reversed by Resveratrol and Cytochrome C Supplementation in Storage”. The Authors performed a series on new experiments using a method of light transmission aggregometry to assess a function of platelets stored at various conditions. Besides, the manuscript was modified according to the suggestions of three reviewers. In my opinion, the paper was significantly improved, however I have one comment.

The Authors used PRP in the light transmission aggregometry (lines 97-98) in the concentration of 500,000 cells/ul. It is high concentrations exceeding both typical concentration of PRP in aggregation studies 200,000-300,000 cells/ul, as well as physiological concentration of platelets in blood. Could the Authors explain a rationale of using this approach?

We thank the reviewer for highlighting this discrepancy from other studies employing PRP. While previous studies report adjusting the platelet count to be tested in PRP to a final count of 200,000-300,000 cells/µL, the Chrono-log model 700 Aggregometer permits counts >250,000 cells/µl while adjusting to establish a stable optical baseline. When we performed an initial analysis at 200,000-300,000 cells/µl, it was not possible to set up an optical baseline without large oscillations that prevented an accurate recording of aggregation. By adjusting the cell count to 500,000 cells/µl a stable baseline was achieved. To address this concern and provide insight for other investigators looking to perform similar studies, comment has been made in the Methods section.

Reviewer 2 Report (Previous Reviewer 3)

Although this work might be an important contribution to current effort
undertaken to develop better storage conditions of platelets, t
his Reviewer still has some critical methodological as well conceptual concerns..

Major points

1. Platelet aggregation: TEG is not considered an appropriate method to assess platelet function. % of platelet aggregation is simply not correct. Please modify throughout the manuscript. What is meant by % of platelet aggregation? Which parameter was determined? Which assay was performed in TEG? Please report in more details.

We agree with the reviewer that the use of TEG does not provide an appropriate measure of platelet function. Based on this comment/recommendation, we have supplemented the data obtained from TEG to include new data measuring platelet aggregation performed by aggregometry using a Chronolog 700 aggregometer. The Methods and Results/figures have been changed accordingly.

REVEIWER: The authors added the aggregometry data but they wrote above “we have supplemented the data obtained by TEG with aggregation performed by aggregometry”. However, in the MS the previous data obtained using the TEG were completely removed, but the description of TEG assay is still present in the Methods section.

2. Is ADP a standard agonist in TEG? Did the authors test other agonist such as thrombin?

As previously described, the TEG data has been supplemented with aggregometry in which we tested ADP at a final concentration of 10μM, and AA at a final concentration of 0.5mM.

REVEIWER: The author added AA as agonist for the aggregation analysis but the description of the results showed in Fig. 1, is not completely correct. In fact, the authors wrote (Pag. 4, line 161-163) “ADP and AA-dependent platelet aggregation decreased during storage in control platelets on day 5, 7, and 10 versus day 1 in both 22°C and 4°C storage (Figure 1A-D)”, but Fig 1D (% A-A aggregation at 4°C) showed clearly not differences at each storage time points.

Moreover, in Fig 1A and C the statistic, day 5, 7, and 10 versus day 1 (as depicted for Fig. 1B) is missing.

3. The authors should use either platelet aggregation assays (Aggregometer) or flow cytometer to assess platelet function.

The TEG data has now been supplemented with aggregometry data.

OK

4. Please report on the volumes and cell concentrations in the final platelet bags.

As recommended, the volumes and cell concentrations have been included in the revised manuscript.

OK

5. Please provide more details on the statistical analyses and explain the symbols in the figure legends.

We apologize for this lack of clarity and have expanded the statistical methods and included the symbol definitions in the figure legends.

REVEIWER: The sentences added (Pag. 4, lines156-158) do not give more information about the methods used to analyzed the data. The authors only explained the symbols used in the figures.

Minor points  

2. Pag 2e, line 52, 53: “Doing so, in 50 conjunction with reducing storage temperature to 4°C, extended platelet aggregation 51 properties ex vivo and maintained mitochondrial and platelet function beyond the stand- 52 ard (5 day) shelf life [13,14].” The verb is missing.

REVEIWER: The verb is still missing. In fact, these sentences are not marked with yellow as edited part of the MS.

4. Page 2, line 74, 75 “Platelet concentrates stored at 22°C were subject to mild agitation 72 (70rpm) in accordance with recommended storage conditions for clinical application. Those stored at 4°C did not undergo agitation because previous reports demonstrate agitation does not improve function of platelets at this temperature [15].”This is might be true but will affect the results (increased aggregation, higher baseline CD62p etc): Please discuss.

We thank the reviewer for highlighting this important point. The Discussion section has been edited accordingly.

REVEIWER: Nothing is written in the Discussion section to clarify this point.

 6. Figure 2, why ROS concentration increased only between day 1 and 5 and then it decreased on day 7 and 10 (for both RT and 4C controls)? I would expect an increase during the all storage time as the author wrote on page 8 line 205. Do you have any explanation?

We thank the reviewer for highlighting this and apologize for the lack of clarity on describing the data. As the reviewer indicates, Figure 2A and B demonstrates a significant increase in ROS measurements on day 5, 7 and 10 versus day 1. Although ROS levels on day 7 and 10 are slightly decreased compared to Day 5, these differences were not was not statistically significant. At present we do not have an experimental explanation for these changes and have highlighted them in the Discussion section as areas for future investigation.

REVEIWER: Nothing is written in the Discussion section about this point.

 8. Figure 3: in each graphic the statistic is completely missing, could you please add it at least in the samples and storage time that you took into consideration? Otherwise is not possible to make any robust conclusion on these data.

We apologize for this lack of clarity. The axis titles depict days of storage and respective temperatures. This has now been included in the figure legend.

REVEIWER: -Statistic is still missing for all graphics reported in this Fig.

REVEIWER: -The axis titles are, x-axis “RCR” and y-axis “Day”, no temperatures are reported as written above.

 10. Why the authors didn’t show 4°C day 1 and RT with Rev or Cyt C?

We apologize for the lack of clarity in these figures. Control day 1 corresponds to platelet samples prior to storage (22°C and 4°C) and the figure has been updated accordingly.

REVEIWER: Day 1 as baseline is now clear but two points are is still unclear looking at Figure 5

I.             Why only samples incubated with Rev and Cyt C stored at 4°C were reported for day 7 and nothing stored at 22°C with Rev and Cty C.

II.            Page 7, line 215-216, what do you mean with “freshly isolated platelets”? platelets before storage or platelet isolated from a random daily donor used as control? Because in Fig. 5 “freshly isolated platelets” are not reported.

 11. Figure 4G why day 1 is reported only for RT and all samples are compared to it even samples stored at 4°C. It is confusing and not totally correct. According to the statistic symbols, it seems that d7, 4°C Cyt C has a higher scoring compared to 4°C control. If it is true then the sentence on page 8 line 182 is not correct (“Cold storage improves platelet structural integrity, effects that are not augmented by cytochrome c or resveratrol supplementation.”).

The reviewer raises an important point. Day 1 represents the baseline (point where platelets were collected prior to storage at 22°C or 4°C storage). Edits have been made on Figure 4. The sentence on page 8 [line 182] concludes that cold stored platelets maintained structural integrity compared to platelets stored at 4°C. However, supplementation with Cyt c and resveratrol at 4°C does demonstrate higher scoring, possibly due to artifacts generated by uptake of Res and Cyt c. To address this the sentence has been revised.

REVEIWER: If day 1 is baseline (prior to storage) why you wrote in the fig. legend “ * vs. day 1 at 22°C”. 

According to your Fig legend the two symbols # and ¶ were obtained comparing the sample not with day 1, as for all previous figures, but to different days of storage and storage conditions (# vs. day 7 at 22°C and ¶ vs. day 4 at 4°C). Why?

Moreover, what you wrote above in your response “The sentence on page 8 [line 182] concludes that cold stored platelets maintained structural integrity compared to platelets stored at 4°C.” make no sense because cold-stored platelets and platelets stored at 4°C are synonymous and you cannot compare them.

Author Response

The authors would like to thank the reviewers for providing their detailed and constructive comments for our resubmitted manuscript studying platelet morphology and function under altered storage conditions. We have now had the opportunity to carefully consider the comments and have addressed them through revisions to the manuscript body and figures. As requested, we have created two versions of the revised manuscript, one in which changes are highlighted and a second clean version. To further assist the Reviewers, we have addressed the points raised herein.

As the reviewers are aware, in the original manuscript we used TEG as an indicator of platelet function. In doing so, the reviewers highlighted that TEG was not an appropriate means of measuring platelet function, and to do so studies should be performed using aggregrometry or flow cytometry. We agreed with these comments and performed the aggregrometry studies as requested. With the resubmission we included both TEG and aggregometry data for comparison. However, as highlighted by the reviewers following this review, this causes confusion in data presentation, interpretation and discussion. To address this, we believe the data is better represented by removing the TEG data completely and only presenting data obtained by aggregrometry.

In addition, as highlighted by the reviewers, the nomenclature used to designate the time from platelet isolation to use was confusing as it suggested the platelets were kept for 1 day prior to experiments beginning. This was not the case and to address this, platelets have been designated Day 0 in the text and figures.

Reviewer 2

N.B. For clarification for the other reviewers, reviewer 2 was kind enough to provide additional feedback to our original responses to concerns they raised. These (original comments/replies) are provided in grey text along with their new comments following the resubmission (red text) and our reply.

Major points

  1. Platelet aggregation: TEG is not considered an appropriate method to assess platelet function. % of platelet aggregation is simply not correct. Please modify throughout the manuscript. What is meant by % of platelet aggregation? Which parameter was determined? Which assay was performed in TEG? Please report in more details.

We agree with the reviewer that the use of TEG does not provide an appropriate measure of platelet function. Based on this comment/recommendation, we have supplemented the data obtained from TEG to include new data measuring platelet aggregation performed by aggregometry using a Chronolog 700 aggregometer. The Methods and Results/figures have been changed accordingly.

  • REVIEWER: The authors added the aggregometry data but they wrote above “we have supplemented the data obtained by TEG with aggregation performed by aggregometry”. However, in the MS the previous data obtained using the TEG were completely removed, but the description of TEG assay is still present in the Methods section.

As highlighted (reply to all reviewers), for greater clarity in data presentation, interpretation, and discussion, the TEG data has been removed from the manuscript and only the aggregometry data is presented.

  1. Is ADP a standard agonist in TEG? Did the authors test other agonist such as thrombin?

As previously described, the TEG data has been supplemented with aggregometry in which we tested ADP at a final concentration of 10μM, and AA at a final concentration of 0.5mM.

  • REVIEWER: The author added AA as agonist for the aggregation analysis but the description of the results showed in Fig. 1, is not completely correct. In fact, the authors wrote (Pag. 4, line 161-163) “ADP and AA-dependent platelet aggregation decreased during storage in control platelets on day 5, 7, and 10 versus day 1 in both 22°C and 4°C storage (Figure 1A-D)”, but Fig 1D (% A-A aggregation at 4°C) showed clearly not differences at each storage time points.
  • Moreover, in Fig 1A and C the statistic, day 5, 7, and 10 versus day 1 (as depicted for Fig. 1B) is missing.

As mentioned, the TEG data has been removed from the manuscript and replaced with aggregometry. In addition, the figures for aggregometry have been updated to include the statistical analyses requested.

  1. The authors should use either platelet aggregation assays (Aggregometer) or flow cytometer to assess platelet function.

The TEG data has now been supplemented with aggregometry data.

- REVIEWER: OK

  1. Please report on the volumes and cell concentrations in the final platelet bags.

As recommended, the volumes and cell concentrations have been included in the revised manuscript.

- REVIEWER: OK

  1. Please provide more details on the statistical analyses and explain the symbols in the figure legends.

We apologize for this lack of clarity and have expanded the statistical methods and included the symbol definitions in the figure legends.

  • REVIEWER: The sentences added (Pag. 4, lines156-158) do not give more information about the methods used to analyze the data. The authors only explained the symbols used in the figures.

We apologize for the confusion (on our part) with replying to the original comment. As requested, we have included additional information for the specific statistical tests performed in the methods and included symbol definitions in the figure legends

Minor points  

  1. Pag 2e, line 52, 53: “Doing so, in conjunction with reducing storage temperature to 4°C, extended platelet aggregation 51 properties ex vivo and maintained mitochondrial and platelet function beyond the standard (5 day) shelf life [13,14].” The verb is missing.

- REVIEWER: The verb is still missing. In fact, these sentences are not marked with yellow as edited part of the MS.

We apologize for this omission. This sentence has been re-written for greater clarity/correct grammar and highlighted as requested.

  1. Page 2, line 74, 75 “Platelet concentrates stored at 22°C were subject to mild agitation 72 (70rpm) in accordance with recommended storage conditions for clinical application. Those stored at 4°C did not undergo agitation because previous reports demonstrate agitation does not improve function of platelets at this temperature [15].” This might be true but will affect the results (increased aggregation, higher baseline CD62p etc): Please discuss.

We thank the reviewer for highlighting this important point. The Discussion section has been edited accordingly.

  • REVIEWER: Nothing is written in the Discussion section to clarify this point.

We apologize for the error made in our original reply to the reviewer. A brief response to this comment was included in the Methods section, but we agree that it is better suited for the Discussion. The Discussion has been updated to reflect the potential impact of not agitating platelets stored at 40C on other parameters.

  1. Figure 2, why ROS concentration increased only between day 1 and 5 and then it decreased on day 7 and 10 (for both RT and 4C controls)? I would expect an increase during all storage time as the author wrote on page 8 line 205. Do you have any explanation?

We thank the reviewer for highlighting this and apologize for the lack of clarity on describing the data. As the reviewer indicates, Figure 2A and B demonstrates a significant increase in ROS measurements on day 5, 7 and 10 versus day 1. Although ROS levels on day 7 and 10 are slightly decreased compared to Day 5, these differences were not statistically significant. At present we do not have an experimental explanation for these changes and have highlighted them in the Discussion section as areas for future investigation.

- REVEIWER: Nothing is written in the Discussion section about this point.

We apologize for the lack of clarity in describing the data and not including this in the Discussion. As the reviewer indicates, Figure 2A & B demonstrates a significant increase in ROS measurements on days 5, 7 and 10 when compared to day 1. Although ROS levels on day 7 and 10 are decreased compared to Day 5, these differences were not statistically different. At present we do not have an experimental explanation for these changes, however, while it is possible due to exhaustion of cellular metabolic function limiting the ability to produce further ROS, that could only be a speculation and have been highlighted in the Discussion section as areas for future investigation.

  1. Figure 3: in each graphic the statistic is completely missing, could you please add it at least in the samples and storage time that you took into consideration? Otherwise is not possible to make any robust conclusion on these data.

We apologize for this lack of clarity. The axis titles depict days of storage and respective temperatures. This has now been included in the figure legend.

- REVEIWER: -Statistic is still missing for all graphics reported in this Fig.

- REVEIWER: -The axis titles are, x-axis “RCR” and y-axis “Day”, no temperatures are reported as written above.

We apologize for misinterpreting the reviewer’s original comment/suggestion. The Figure legends have been edited to include definitions of the significance markers used. In addition, we have changed the format of the graphs to bar charts (from line graphs) for greater ease in interpretation and clarified both the axis labels and the storage temperatures for each data set.

  1. Why the authors didn’t show 4°C day 1 and RT with Rev or Cyt C?

We apologize for the lack of clarity in these figures. Control day 1 corresponds to platelet samples prior to storage (22°C and 4°C) and the figure has been updated accordingly.

- REVEIWER: Day 1 as baseline is now clear but two points are is still unclear looking at Figure 5

  1. Why only samples incubated with Rev and Cyt C stored at 4°C were reported for day 7 and nothing stored at 22°C with Rev and Cty C.
  2. Page 7, line 215-216, what do you mean with “freshly isolated platelets”? platelets before storage or platelet isolated from a random daily donor used as control? Because in Fig. 5 “freshly isolated platelets” are not reported.

Studies have been performed and data for samples stored at 22°C with Res and Cyt c have been included in the figure as requested.

As previously noted in response to all reviewers, there was confusion that arose in describing platelet collection and storage. To clarify, the baseline is now designated day 0 and we provide a more complete description regarding collection times and experimental start.

  1. Figure 4G why day 1 is reported only for RT and all samples are compared to it even samples stored at 4°C. It is confusing and not totally correct. According to the statistic symbols, it seems that d7, 4°C Cyt C has a higher scoring compared to 4°C control. If it is true then the sentence on page 8 line 182 is not correct (“Cold storage improves platelet structural integrity, effects that are not augmented by cytochrome c or resveratrol supplementation.”).

The reviewer raises an important point. Day 1 represents the baseline (point where platelets were collected prior to storage at 22°C or 4°C storage). Edits have been made on Figure 4. The sentence on page 8 [line 182] concludes that cold stored platelets maintained structural integrity compared to platelets stored at 4°C. However, supplementation with Cyt c and resveratrol at 4°C does demonstrate higher scoring, possibly due to artifacts generated by uptake of Res and Cyt c. To address this the sentence has been revised.

- REVEIWER: If day 1 is baseline (prior to storage) why you wrote in the fig. legend “ * vs. day 1 at 22°C”. 

According to your Fig legend the two symbols # and ¶ were obtained comparing the sample not with day 1, as for all previous figures, but to different days of storage and storage conditions (# vs. day 7 at 22°C and ¶ vs. day 4 at 4°C). Why?

Moreover, what you wrote above in your response “The sentence on page 8 [line 182] concludes that cold stored platelets maintained structural integrity compared to platelets stored at 4°C.” make no sense because cold-stored platelets and platelets stored at 4°C are synonymous and you cannot compare them.

We apologize for the lack of clarity with the figures and corresponding legends. Control day 1 corresponds to platelet samples prior to storage (22°C and 4°C). We appreciate the confusion caused by the labeling of these graphs and have revised the manuscript and figures to indicate “day 1” samples are better described as “day 0 (baseline)”. In addition, the figures/axis and figure legends have been updated for greater clarity and to indicate the storage temperatures with appropriate description of the comparison groups denoted by symbols.

Finally, we thank the reviewer for highlighting the mistake regarding comparing cold storage to 40C. As the reviewer indicates, these are the same conditions, and there was an error in this statement. This has been corrected in the revised manuscript.

Reviewer 3 Report (New Reviewer)

In this work, the authors study in vitro effects of resveratrol and cytochrome c on platelet function (ADP induced aggregation, ROS production, mitochondrial function and morphology).

The improvement of platelet storage protocols is a rerelevant  clinical question and the  manuscript is clearly presented. However, some major points need to be addressed before further consideration:

1/Title is unapropriate: according to the main results of the paper, Res and cytc do not reverse the alteration of platelet mitochondrial function. 

2/previous studies on resveratrol on platelets including one by the authors themselves should be cited: PMID 29538230 and 26683619.

3/page 2 lines 64 to 66, I don't understand the sentence.

4/In each figure legend it is stated that 5 independent experiments were realized. However, in Materials and Methods section it is stated that only 4 sets of pooled PRP were prepared. Could the authors explain how it is possible ?

5/ Platelet parameters were analysed on days 1, 5, 7 and 10. It would have been informative to have the baseline values at day 0.

6/Page 3 line 119 and 120. Platelet counting method should be cited earlier in the manuscript, page 2 line 85.

7/Page 4 line 164: res and Cyt c do not preserve platelet function on d7 and 10. They just limit ADP aggregation decline.

8/Figure 1B: please mention that res and Cyt c have an effect on ADP aggregation at 4°C in the legend.

9/Figure 1 C and D: it is unclear if AA aggregation declined at 22°C and especially at 4°C (*p-values of control conditions versus d1 should be specified).

10/Page 5, please rename the paragraph title thromboelastography, not platelet aggregation.

11/Figure 2: A and B are missing on the figure. Figure 2B: Res and Cyt c seem to increase ADP-TEG compared to control at d1, especially at d7. Can the authors comment on this result ?

12/Page 6 line 201, it should be stated here that although RCR is maintained >1, there is no statistical difference between control and Res or Cyt c conditions at d5.

12/Page 8, line 234, "abrogated" is unappropriate.

Author Response

The authors would like to thank the reviewers for providing their detailed and constructive comments for our resubmitted manuscript studying platelet morphology and function under altered storage conditions. We have now had the opportunity to carefully consider the comments and have addressed them through revisions to the manuscript body and figures. As requested, we have created two versions of the revised manuscript, one in which changes are highlighted and a second clean version. To further assist the Reviewers, we have addressed the points raised herein.

As the reviewers are aware, in the original manuscript we used TEG as an indicator of platelet function. In doing so, the reviewers highlighted that TEG was not an appropriate means of measuring platelet function, and to do so studies should be performed using aggregrometry or flow cytometry. We agreed with these comments and performed the aggregrometry studies as requested. With the resubmission we included both TEG and aggregometry data for comparison. However, as highlighted by the reviewers following this review, this causes confusion in data presentation, interpretation and discussion. To address this, we believe the data is better represented by removing the TEG data completely and only presenting data obtained by aggregrometry.

In addition, as highlighted by the reviewers, the nomenclature used to designate the time from platelet isolation to use was confusing as it suggested the platelets were kept for 1 day prior to experiments beginning. This was not the case and to address this, platelets have been designated Day 0 in the text and figures.

Reviewer 3

In this work, the authors study in vitro effects of resveratrol and cytochrome c on platelet function (ADP induced aggregation, ROS production, mitochondrial function and morphology).

The improvement of platelet storage protocols is a relevant clinical question, and the manuscript is clearly presented. However, some major points need to be addressed before further consideration:

  1. Title is inappropriate: according to the main results of the paper, Res and cyt c do not reverse the alteration of platelet mitochondrial function. 

We agree with the reviewer and have changed the title of the manuscript accordingly.

  1. Previous studies on resveratrol on platelets including one by the authors themselves should be cited: PMID 29538230 and 26683619.

As suggested by the reviewer, both references have been included in the revision.

  1. page 2 lines 64 to 66, I don't understand the sentence.

We apologize for the lack of clarity with this sentence and have edited it as requested.

  1. In each figure legend it is stated that 5 independent experiments were realized. However, in Materials and Methods section it is stated that only 4 sets of pooled PRP were prepared. Could the authors explain how it is possible?

We thank the reviewer for highlighting this error and for our lack of clarity in providing adequate descriptions of the collection and allocation of platelets to the respective experimental groups. Briefly, following IRB approval and having obtained informed consent, platelet samples were collected from 20 individual donors at The Community Blood Center of the Carolinas. From these individual samples 4 sets of pooled platelet samples were obtained, each pooled platelet sample consisting of platelets from 5 different donors.  Each pooled platelet sample was subdivided into 6 units and designated to an experimental group (Control; Res; Cyt C) at 4°C or room temperature. At each time point designated an aliquot was removed and assayed in triplicate. Thus, the experimental N for each group should have been listed as N=4 (ie 20 donors, 5 donors pooled per experimental group). This has been clarified in the Methods section and the manuscript updated accordingly.

  1. Platelet parameters were analyzed on days 1, 5, 7 and 10. It would have been informative to have the baseline values at day 0.

We agree with the reviewer that, based on the experimental groups, day 0 is a better descriptor than day 1 (see also comments to all reviewers). The manuscript has been edited accordingly.

6.Page 3 line 119 and 120. Platelet counting method should be cited earlier in the manuscript, page 2 line 85.

We agree with the reviewer and this edit has been made to include the recommended changes in the Methods section.

  1. Page 4 line 164: res and Cyt c do not preserve platelet function on d7 and 10. They just limit ADP aggregation decline.

We thank the reviewer for raising this pertinent point. The manuscript has been edited to reflect this.

  1. Figure 1B: please mention that res and Cyt c have an effect on ADP aggregation at 4°C in the legend.

& 9. Figure 1 C and D: it is unclear if AA aggregation declined at 22°C and especially at 4°C (*p-values of control conditions versus d1 should be specified).

& 10. Page 5, please rename the paragraph title thromboelastography, not platelet aggregation.

& 11. Figure 2: A and B are missing on the figure. Figure 2B: Res and Cyt c seem to increase ADP-TEG compared to control at d1, especially at d7. Can the authors comment on this result?

In light of the suggestions and comments received from previous reviews, we have decided to remove the TEG data from the manuscript altogether as we feel this confuses the presentation and interpretation of the data obtained with the performing studies using the aggregometer. 

  1. Page 6 line 201, it should be stated here that although RCR is maintained >1, there is no statistical difference between control and Res or Cyt c conditions at d5.

We agree with the reviewer and this statement has been included in the revision.

  1. Page 8, line 234, “abrogated” is unappropriate.

We agree with the reviewer and this statement has been amended.

Round 2

Reviewer 1 Report (Previous Reviewer 1)

I have no further comments.

Reviewer 3 Report (New Reviewer)

The authors satisfactorilly answered to all my comments.

This manuscript is a resubmission of an earlier submission. The following is a list of the peer review reports and author responses from that submission.

Round 1

Reviewer 1 Report

In the original article entitled: „Altered Platelet Morphology and Mitochondrial Function is reversed by Resveratrol and Cytochrome c Supplementation in Storage”, by Dr Ekaney et al., the Authors analysed the effects of natural oxidant resveratrol as well as Cytochrome C on preservation of platelet function in stored donated platelets. I think that the study is interesting, however I have several comments.

Major comment

The Authors selected resveratrol, a compound belonging to plant polyphenols having antioxidant activity, however there is a number of other polyphenolic compound having the same properties (e.g. quercetin). What was a rationale to select resveratrol? Are the Authors planning to screen other antioxidants in the future?

Minor comments

  1. In the Material and Methods section there is no description of platelet aggregation procedure.
  2. No ADP concentration used as an agonist in the platelet aggregation is shown.
  3. There is no information whether the effect of Res and Cyt C was measured also at 22 oC (Fig 1 presents only effects of Res and Cyt C at 4oC).
  4. In the legend of Fig 1., the Authors should explain “*” and “#”, show the number of experiments, give the name of statistical test (it is not enough to describe it in the Statistical analysis).

Reviewer 2 Report

The manuscript by Ekaney et al deals with respiratory and oxydative cellular lesions during storage of platelets in plasma  before being transfused to patients.

the study is well explained and experiments are of interest. The results in term of alteration in platelets :  function (aggregation) , structure (electronic microscopy), oxydative stress (ROS / RCR)  are  demonstrative; partial reversion of ROS production is shown resveratrol and cytochrome c supplementation and storage at 4 °C but no in term of RCR.

However, there are some weakness

Major points:

Point 1 : choice of the model

Only platelet prepared and stored in plasma ,: unless it it a standard classic preparation, use of such platelets for transfusion is decreasing since the inactivation of pathogen process had  demonstrated its saftey ,and its clinical efficiency.  Platelet prepared with pathogen inactivation (UV treatment based methods) process already allows a 7 days storage duration . Such platelets are prepared not in 100% plasma but in additive solution ( about 30 % residual plasma).   So the results shown are of poor impact in blood banking practice.

Plasma platelets are not the standard of many blood banks  ,either by national policies (mainly in Europe)  either by local decicion  of blood banks (other countries i e North America). The authors must provide comparative data on various platelets preparation for human use.

Point 2 : as consequence of point 1 since the aim of the study is to provide arguments to increase the duration of storage , are the presented  methods  to counteract the oxydative stress are compatible with human use and safety ?  The authors provided no convincing data for their choice.

Point 3: the role of  oxydative stress   (ROS and RCR )  in platelet storage is not an original finding as the authors wrote, so unless experiments are clearly exposed and the results supported by experiments in term of ROS  reversion, the lack of significance in term of RCR  reversion appears  as a weakness. 

Point 4 : minor 

The vast majority of platelets transfusion is for medicine (heamatology/oncology)  for prophylaxis or bleeding control, and  not for trauma and surgery support.  Improvment in the choice of references is needed.

Reviewer 3 Report

Dr Ekaney and colleagues report in this paper on the impact of Resveratrol and Cytochrome c on altered platelet morphology and mitochondrial function. The main finding of this study is that Declining platelet function and increased intracellular ROS can be abrogated by Res and Cyt c.

Although this work might be an important contribution to current effort undertaken to develop better storage conditions of platelets, this Reviewer has some critical methodological as well as conceptual concerns.

Major issues:

  1. Platelet aggregation: TEG is not considered an appropriate method to assess platelet function. % of platelet aggregation is simply not correct. Please modify throughout the manuscript. What is meant by % of platelet aggregation? Which parameter was determined? Which assay was performed in TEG? Please report in more details.
  2. Is ADP a standard agonist in TEG? Did the authors test other agonist such as thrombin?
  3. The authors should use either platelet aggregation assays (Aggregometer) or flow cytometer to assess platelet function.
  4. Please report on the volumes and cell concentrations in the final platelet bags.
  5. Please provide more details on the statistical analyses and explain the symbols in the figure legends

Minor points:

- Abstract line 17, add the days of storage (1, 5, 7 and 10)

-Pag 2, line 52, 53 : “Doing so, in 50 conjunction with reducing storage temperature to 4°C, extended platelet aggregation 51 properties ex vivo and maintained mitochondrial and platelet function beyond the stand- 52 ard (5 day) shelf life [13,14].” The verb is missing.

-Pag 2, line 65,66 “Whole blood-derived plate- 65 lets were isolated and pooled into 4 sets (5 donors/set), each of which). Each were then 66 divided into 6 individual platelet-storage bags (Haemonetics, Braintree, MA)” some editing mistakes, please correct.

-Pag 2, line 74, 75 “Platelet concentrates stored at 22°C were subject to mild agitation 72 (70rpm) in accordance with recommended storage conditions for clinical application. Those stored at 4°C did not undergo agitation because previous reports demonstrate agitation does not improve function of platelets at this temperature [15].”.

This is might be true but will affect the results (increased aggregation, higher baseline CD62p etc): Please discuss.

-Pag 4, Figure 1

a)Why in Fig 1B the authors didn’t show the samples: RT with and without Rev and Cyt C, as you did for figure 2?

 -Figure 2, why ROS concentration increased only between day 1 and 5 and then it decreased on day 7 and 10 (for both RT and 4C controls)? I would expect an increase during the all storage time as the author wrote on page 8 line 205. Do you have any explanation?

-Legend of figure 2: “Res or Cyt c significantly decreased ROS production (#p<0.05, versus day 1). B). Intracellular ROS production was significantly increased in after 5, 7 and 10 days storage at 4°C (*p<0.05 versus day 1), and Res or Cyt c significantly decreased ROS production (#p<0.05, versus day 1).”

If the authors want to investigate the impact of Rev and Cyt C they should compare them not with day 1 control but with the control sample of the corresponding day. For example day 5 control vs. Rev

-Figure 3: in each graphic the statistic is completely missing, could you please add it at least in the samples and storage time that you took into consideration? Otherwise is not possible to make any robust conclusion on these data.

-Figure 4: add the name of the samples and the corrisponding days for each picture.

Why the authors didn’t show 4°C day 1 and RT with Rev or Cyt C?

-Figure 4G why day 1 is reported only for RT and all samples are compared to it even samples stored at 4°C. It is confusing and not totally correct. According to the statistic symbols, it seems that d7, 4°C Cyt C has a higher scoring compared to 4°C control. If it is true then the sentence on pag 8 line 182 is not correct (“Cold storage improves platelet structural integrity, effects that are not augmented by cytochrome c or resveratrol supplementation.”)

-Page 8 line 191 “Here, we observed a decline in platelet aggregation during storage to be associated with increased ROS levels throughout storage time” this is not totally correct in fact an increase of ROS level was observed only between day 1 and 5 while starting from day 7 on the levels are decreasing.

-page 8, line 222-230 “Consistent with previous findings, our study reports that loss in mitochondrial respiratory function could be mediated through CI and CIV. For example, Perales et al. report a dramatic decrease in platelet mitochondrial respiration for CI and CIV within the first 2 days of storage under standard (22°C) conditions. [6] Similarly, Diab et al. report decreaing CIV activity during platelet storage [24]. This decreased respiratory capacity likely  contributes to the decline in platelet aggregation, as platelet activation requires the energy  production provided by mitochondrial respiration [7]. In addition, we assessed the effects of platelet storage on respiratory capacity of CII, finding that CII is not affected. This suggests that CII is not a target for interventions to prevent platelet dysfunction during storage.”

Why did authors discuss separately CI and CII affects when they used it together? How can you distinguish their effects?

-Page 8 line 218-220 “Similar to ROS production, RCR was not affected by temperature in our study. This is in contrast with previous studies that report platelets stored at 4°C conserved mitochondrial respiration and maximal mitochondrial utilization better than room temperature 220 platelets at day 7 of storage [23].” Do the authors have any explanation?